# Structural gaps of water resources knowledge in global river basins

Shuanglei Wu[1], Yongping Wei[1,*], Xuemei Wang[2]

[1]: School of Earth and Environmental Sciences, the University of Queensland, Brisbane, 4072, Australia.

[2]: University Library, Southwest University, Chongqing, 400715, China.

*Correspondence to*: Yongping Wei (yongping.wei@uq.edu.au)

**Abstract:** The stationarity of hydrological systems is dead in the era of the Anthropocene. Has our hydrological/water resources knowledge well transformed to address this change? By using publications indexed in the Web of Science database since 1900, we aim to investigate the global development of water resources knowledge at river basin scale with a system approach, of which water resources knowledge development in a river basin is defined as a complex system involving the co-evolutionary dynamics of scientific disciplines and management issues. It is found that: 1) legacy-driven water resources knowledge structures have consistently dominated most of the highly researched river basins in the world, while innovation-driven structures are identified in the river basins receiving increasing research publications in the recent period; 2) the management issues addressed by legacy-driven river basin studies are increasingly homogenised, while wider range of emerging issues are considered by innovation-driven river basin studies; and 3) cross-disciplinary collaborations have remained largely unchanged and collaborations with social sciences have been very limited. It is concluded that the stationarity of water resources knowledge structure persists. A structural shift of water resources knowledge development is urgently needed to cope with the rapidly changing hydrological systems and associated management issues, and opportunities for such shift exist in those less researched, but globally distributed, innovation-driven river basins.

## 1. Introduction

Humans have made substantial impacts on various Earth system cycles, marking the transition of our planet into the Anthropocene (Crutzen, 2002; Crutzen and Stoermer, 2000). This has been powered by the development of science and technology in particular since the Industrial and Scientific Revolutions (Lubell and Morrison, 2021; Steffen et al., 2011; Lewis and Maslin, 2015). Thus, rethinking scientific development in the Anthropocene is crucial for our future survival. The hydrological cycle is a central component of the Earth system and it is widely recognized that the stationarity of the hydrological system is dead as a result of human impacts (Milly et al., 2008; Ajami et al., 2017; Birkinshaw et al., 2014). Has our hydrological and water resources knowledge well transformed to support water resources management in the changing conditions?

Knowledge is typically recognised as a system. Scientific knowledge represents "ordered knowledge of phenomena and the rational study of the relations between the concepts in which those phenomena are expressed" (Dampier, 1944). Recently, scientific knowledge is increasingly recognised as a complex and dynamic system network in which scientists, disciplines and phenomena to be "weaved together into an overarching scientific fabric" (Latour, 1987). The complex interdependencies in the fabric are considered as the structure of the knowledge system (Shi et al., 2015; Coccia, 2020), and that the functionality of the complex system depends on its structure (Von Bertalanffy, 1968; Huttenhower et al., 2012; Sayles and Baggio, 2017). The structure of a disciplinary knowledge system is often analysed in two primary ways (Cheng et al., 2020). First, discipline experts qualitatively review and assess theoretical advances, methods development, and key challenges in the field based on their research experiences and professional knowledge (e.g., Savenije et al., 2014; Mcmillan et al., 2016; Sivapalan, 2018). Second, systemic bibliometric studies are conducted to quantitatively investigate the structure of disciplinary knowledge and reveal the interactions among major research topics (e.g., Zare et al., 2017; Zeng et al., 2017). The latter is often used as a

great complement to those findings from professional knowledge and research experience and helps identify the potential knowledge gaps from the structural perspective (Cheng et al., 2020).

Since its existence particularly in the past decades, the development of hydrological/water resources (here after called as water resources) knowledge, under great support from the IAHS (International Association of Hydrological Sciences), the IHP (Intergovernmental Hydrological Programme) and other initiatives, has extended our understanding from empirical engineering designs to a system of sciences that integrates knowledge from chemistry, physics, geology, and ecology (Montanari et al., 2015; Sivapalan, 2018). More recently, there have been increasing interests to integrate findings from sociology, economics, law, history and psychology to address the challenges posed by the increasingly intertwined human-water relationships under climate change (Yu et al., 2020; Di Baldassarre et al., 2019; Savenije et al., 2014). In addition, knowledge has been developed in various ways in different river basins, influenced by interactive dynamics between scientific disciplines engaged and the management issues emerged (e.g., Bouleau, 2014). However, there has been no systemic survey of how these different disciplines have inter-connectedly contributed to the fundamental understanding of river basins (Ison and Wei, 2017).

This study aims to investigate the development of water resources knowledge structure at river basin scale since 1900. We define water resources knowledge development in a river basin as a complex system involving the co-evolutionary dynamics of scientific disciplines and management issues and it is a sub-system of the entire knowledge system covering all scientific disciplines. The complex network system approach is adopted, and the Web of Science database is used as the data source. Specifically, we investigate: 1) the evolution of publications in the water resources discipline; 2) the evolution of the water resources disciplinary structure; 3) links between the disciplinary structure and the management issues; and 4) collaborations of the water resources discipline with other disciplines. It is expected that key findings from this study will complement the knowledge gaps identified by professional knowledge and research experience from a structural perspective, and contribute to the transformation of water resources knowledge in the Anthropocene.

## 2. Methods and Data

### 2.1. Defining the structure of water resources knowledge system

We define the knowledge development in a river basin as a complex system involving scientific disciplines and management issues, each of which have their respective evolutionary dynamics (Von Bertalanffy, 1968; Wu et al., 2020). Network analysis, which can simplify the real systems while preserving the essential information of their interactive structures that lead to the emergent of complex phenomena (Zeng et al., 2017), was used to investigate development of the water resources knowledge structure.

We use two basic network indicators to represent the knowledge structure: centrality (defined as "degree" in network analysis) and diversity (defined as "closeness" in network analysis) (Wasserman and Faust, 1994; Borgatti, 2005). Centrality measures the number of connections a node has in a knowledge network system. It reflects the level of knowledge concentration: the greater the centrality, the more connected a discipline is with other disciplines in the network. Diversity measures the inverse sum of connecting distances to all other nodes. It expresses the extent to which a node is isolated within the knowledge system: the greater the diversity, the fewer extended connections a discipline has and thus forming more confined small groups in the network. Empirical analyses have demonstrated that centralised knowledge structures facilitate dissemination of existing knowledge, whereas isolated structure can increase adaptivity to different disciplinary knowledge and facilitate radical innovations to knowledge development (Bodin and Prell, 2011; Foray, 2018; Schot and Geels, 2008). Based on the value differences of the centrality and diversity indicators, four types of knowledge structures can be defined (Figure 1). They are:

- Ideal structure with high centrality and high diversity. With this structure, the river basin should have high research intensities in core disciplines to provide solid theoretical foundations, while at the same time have sufficient cross-

disciplinary collaborations to ensure knowledge innovations to address unexpected, emerging river basin management challenges;

- Innovation-driven structure with high diversity but low centrality, which could have a risk of discipline hollowing-out (marginalization of influences from core disciplines). For the river basin with this structure, the connection with core disciplines (centrality) should be strengthened;

• Legacy-driven structure with high centrality but low diversity, which could discourage knowledge innovation. For the river basin with this structure, the cross-disciplinary collaborations (diversity) should be strengthened to increase the potential of knowledge pattern transformation against emerging management challenges; and

- Under-developed structure with low centrality and low diversity, indicating that the knowledge development is still at its early stage and the knowledge system should be strengthened comprehensively for balanced development.

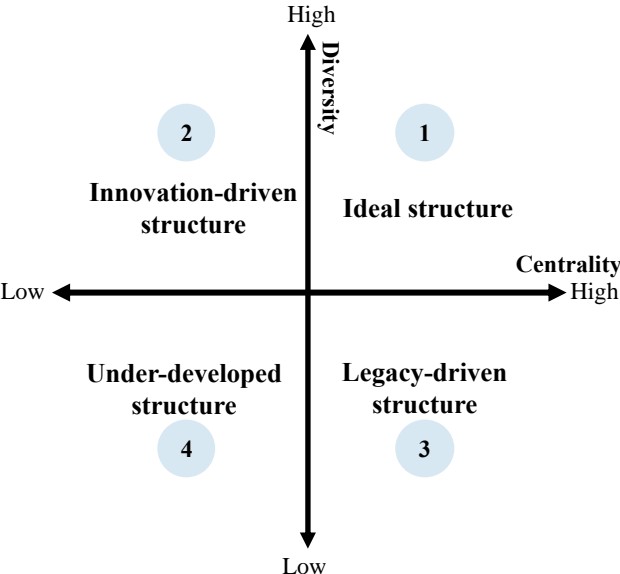

**Figure 1 Definition of four knowledge structures based on their structural indicators.**

By grouping the global river basins based on their knowledge structure and tracking their changes in time, we can identify the structural distribution and the evolutional patterns of global river basin knowledge. By linking these patterns to the management issues of focus, we can empirically identify what type of knowledge structure is more often used to solve what

management issues. Complemented by the understanding of collaborations of water resources discipline with other disciplines, these analyses can provide insights into the transformation of water resources knowledge and assist the strategic design and planning of future research from the structural perspective.

**2.2 Data source**

This study used peer-reviewed publications indexed in the Web of Science (WoS) as the data source. Scientific publications

provide objective documentations of knowledge development, and online academic publication database allow the patterns of knowledge development for different disciplines to be explored (e.g., Xu et al., 2018; Rousseau et al., 2019). As one of the largest academic databases available since 1900, the WoS archives over 12,000 international and regional journals into five major categories: Arts & Humanities, Life Sciences & Biomedicine, Physical Sciences, Social Sciences, and Technology (Engineering), totalling 254 disciplines (Clarivate Analytics, 2018) (refer to Appendix A for details on disciplines grouped

under each category). Water resources is one of these disciplines with a specific focus on water-related studies and covers the major journals in this field (e.g., *Hydrology and Earth System Science, Water Resources Research*, *Journal of Hydrology, Water Research, and Desalination*) (Clarivate Analytics, 2020).

We chose river basin as the spatial unit for analysis as it represents the territorial unit of water cycle linking to other cycles of

the Earth system (e.g., nutrients, energy, and carbon), and is commonly adopted by researchers to understand the integrated impacts of water use, land use and environmental management (Warner et al., 2008; Newson, 2008). We collected the relevant journal articles in WoS by searching for "drainage basin" OR "river basin" OR "valley" OR "hydrographic basin" OR "watershed" OR "catchment" OR "river" OR "wetland" in the Titles, Abstracts and Keywords sections of publications from 1900 to 2017. Firstly, according to the journals in which the retrieved publications were published, each publication was assigned to a discipline. We then merged those publications focusing smaller spatial units (e.g., sub-catchment, or wetland or lake) into the river basin which they are affiliated with, and removed all duplicate publications. Following that, the most researched 100 river basins which covered a majority of the total publications on river basins were selected. After we removed those river basins with incomplete data, a total of 95 river basins were finally used in further analysis (Figure 2).

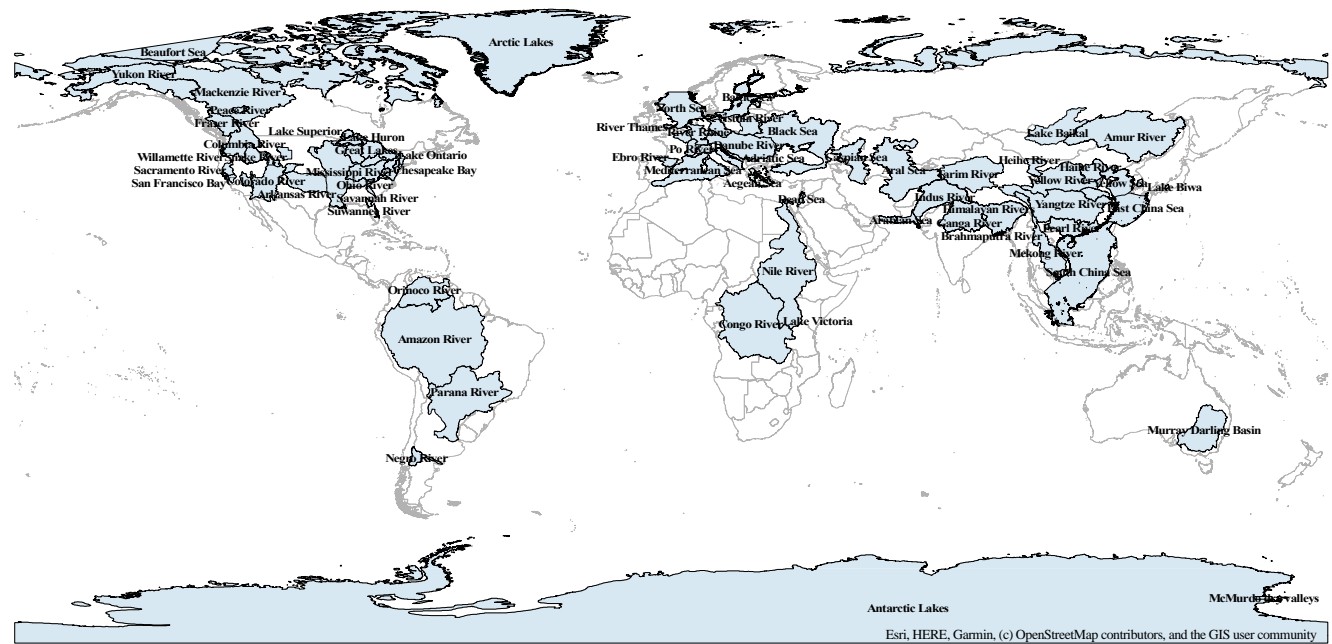

**Figure 2 The spatial locations of the 95 river basins used for analysis.**

**2.3 Key words analysis**

Key words have been widely used to express the research topics of articles and are considered as basic elements in understanding the content and structure of disciplinary knowledge (Khasseh et al., 2017; Cheng et al., 2020). In this study, the management issues in each publication were represented by the key words extracted using the Natural Language Processing (NLP) module in the Derwent Data Analyzer (https://clarivate.com/derwent/zh-hans/solutions/derwent-data-analyzer-automated-ip-intelligence/) from the Titles, Abstracts and Keywords sections of the publication rather than only from the Keywords section to ensure sufficient representation of the issues (Rebholz-Schuhmann et al., 2012). After duplicates, special characters and meaningless stop words were removed, these key words were stemmed and ranked based on their Term Frequency-Inverse Document Frequency (TF-IDF). TF-IDF was calculated to give higher weights to key words with a high appearance frequency in its corresponding section and a low overall appearance frequency in the entire text collection to avoid a bias towards general terms and grasp the newly appeared key words (Xiong et al., 2014).

These computer-mined key words were then grouped manually into the management issues that broadly represent major clusters of river basin management concerns, including: Agricultural irrigation; Climate variability and change; Droughts and floods; Ecological degradation and restoration; Erosion and sedimentation; Surface water and groundwater management; Water pollution and treatment; Water policy/regulation; and Others (not elsewhere classified) (refer to Appendix B for more details about the identified management issue groups and example key words for each group). These nine management issue groups were determined with our data for the whole study period based on several commonly used water thesauri (Ayllón et

al., 2018; Xiong et al., 2016; Wei et al., 2015). It should also be noted that one article may have multiple management issues, and each issue was counted equally. To ensure consistency of key words grouping, two independent coders were asked to group the key words with any ambiguity thoroughly discussed. In particular, the newly appeared key words in each temporal period were carefully examined to reflect the evolution of management issues.

## 2.4 Knowledge networks analysis

The knowledge networks were established based on the co-occurrence principal (Callon et al., 1983). Two disciplines were connected if they were linked to the same issue in an article; and two management issues were connected if they appeared in the same article (Borgatti and Everett, 1997; Borgatti, 2009). These connections were then used to establish a disciplinary network and an issue network respectively for each river basin. The collaborations of the water resources discipline with other disciplines were derived from the disciplinary network.

The disciplinary and issue networks were constructed and the centrality and diversity of the water resources discipline for each river basin were calculated according to the definitions given above using the "igraph" package in R (https://igraph.org/r/) (refer to Appendix C for detailed formulae). To ensure that the river basins classified within the same type of knowledge structure represent similar structural characteristics (centrality and diversity values), the agglomerative hierarchical clustering (AHC) using the "factoextra" package in R (https://cran.r-project.org/web/packages/factoextra/index.html) was conducted to group the river basins. The clustering was performed based on the Euclidean distances and the Ward's agglomerative criterion (Murtagh and Legendre, 2014) for the normalised degree and closeness values (between 0 and 1). The number of groups was determined while the sum of square errors between different groups were maximized and the errors within groups were minimised. Based on the differences of centrality and diversity values, the clustered river basins were then grouped into different knowledge structures defined above (Figure 1).

## 2.5 Temporal periods division

We divided the whole study time into temporal periods to analyse the evolution of water resources knowledge structure and its links with management issues, and collaboration of water resources with other disciplines. The temporal periods were identified using the nonparametric change point detection method in the "changepoint" package in R (https://cran.r-project.org/web/packages/changepoint/index.html). It calculates the abrupt changes in mean and variances of the total number of articles published in time. The change point detection rather than the trend detection method (e.g., Mann-Kendall test) was used because it focuses on identifying the abrupt changes of publications rather than determining its increasing/decreasing trend in time (Jaiswal et al., 2015; Killick et al., 2012).

## 3. Results

### 3.1 Temporal and spatial distribution of the water resources publications by management issues

The earliest publication on water resources for the 95 mostly published river basins was in 1970, and accumulated to a total of 9128 publications in 2017. As shown in Figure 3a, three development periods were identified. Before 1993, the number of articles published annually were limited (fewer than 250 publications), with the top three management issues being water pollution and treatment (64 publications), surface water and groundwater management (48), and sedimentation and erosion (28). Annual publications began to take off since the 1990s, with an increment of about 10 times. During this second period (1994 – 2005), water pollution and treatment (626) continued to be the focus of studies in these rivers, followed by surface water and groundwater management (388) and water policy (257). Articles on water resources continued to increase during the most recent period (2006 – 2017), although the rate has slowed down (3 times from the previous period). Surface water and groundwater management (1610) and water pollution and treatment (1228) continued to be the centre of issues focussed,

with studies on water policy, climate variability and change, sedimentation and erosion, and ecological degradation and restoration gaining momentums (each with over 550 publications). These management issues particularly those emerging in different temporal periods were consistent with those identified by the mainstream hydrological and water resources communities (e.g., the IAHS's scientific decades: https://iahs.info/ and the IHP's research phases: https://en.unesco.org/themes/water-security/hydrology/IHP-VIII-water-security) (Sivapalan, 2018; Mccurley and Jawitz, 2017; Cudennec et al., 2015).

The spatial distributions of publications on the water resources discipline indicated great diversity among river basins around the globe (Figure 3b). River basins located in the North America and southeast Asia had most publications in all time. The top five are the Yellow River, the Yangtze River, the Mississippi River, the Murray-Darling River, and the Colorado River. Different research preferences were also demonstrated in different river basins. For example, the Yellow River and the Yangtze River received the most focus on surface water and groundwater management, whereas research on the Mississippi River focused on water pollution and treatment and the Murray-Darling River on water policy. Among all river basins, over 38% received most publications on water pollution and treatment issue, 53% of which were located in North America. Over 28% rivers focused on the surface water and groundwater management issue, 46% of which were located in Asia. River basins in Europe (54%) were also most focused on water pollution and treatment. Among the limited number of rivers in South America, Africa, Antarctica and Oceania identified (12% of 95 rivers), the focuses were spread across surface water and groundwater management, ecological degradation and restoration, and water policy.

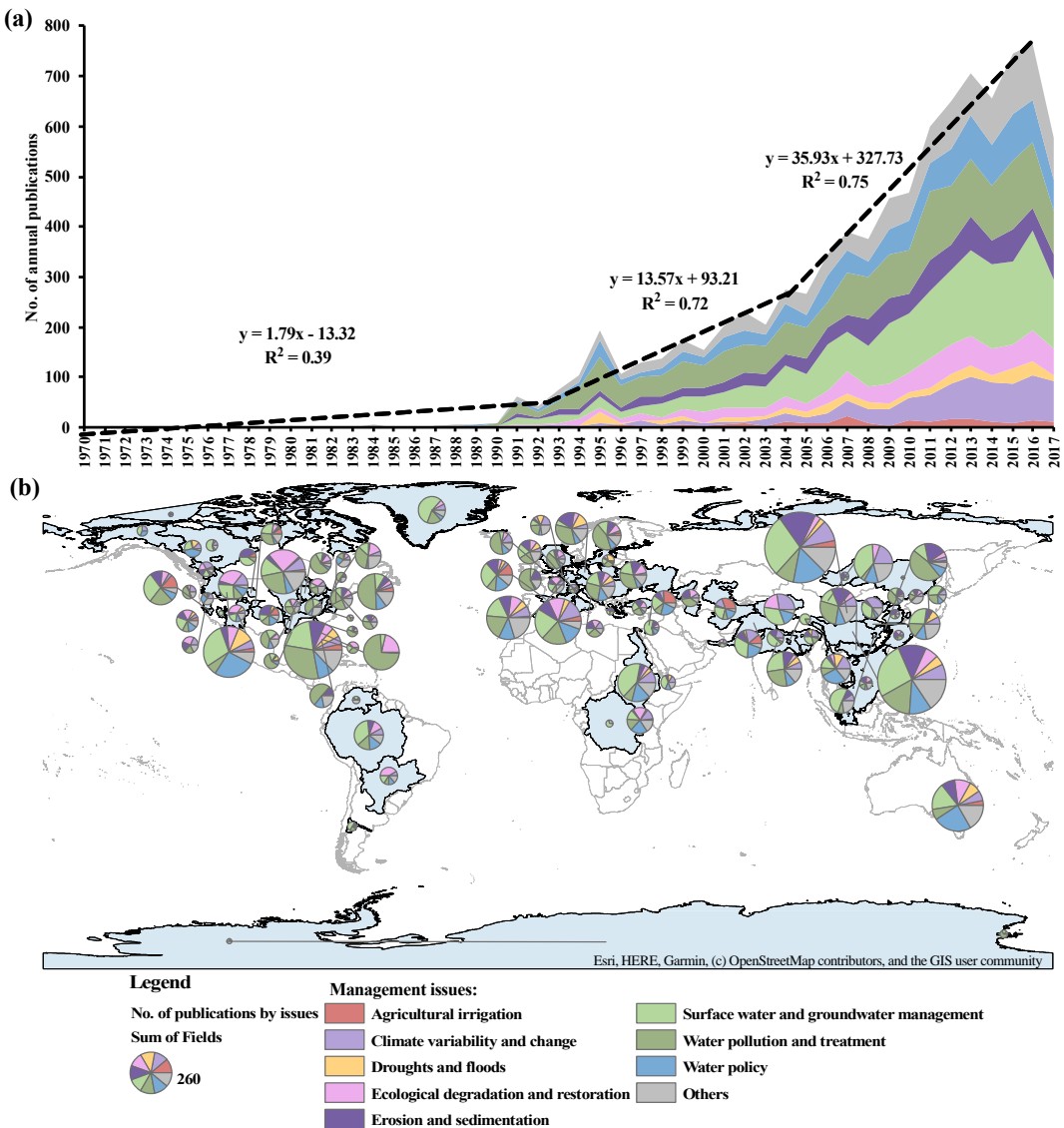

Note: "Other" group contains key words on specific water bodies and/or general terms which could not be grouped in any of the other issues.

**Figure 3 (a) The temporal development of annual publications on water resources, decomposed by the management issues; and (b) the spatial distribution of total publications on water resources, decomposed by the management issues.**

### 3.2 Structural development of the water resources discipline

The knowledge structure of the water resources discipline in each river basin varied in time (Figure 4). During the 1970 – 1993 periods, there were 64 rivers identified as the under-developed structure, spanning across a wide range of spatial regions especially in Asia, Africa, Europe, and some parts of South America and North America. Another 28 rivers were identified to have legacy-driven structures (78% of total publications), indicating centralised development on water resources knowledge in these river basins. They were mainly located in North America and Europe, including the Mississippi River, the Great Lakes, the Mediterranean Sea, and the River Rhine; and some major river basins in Australia, Africa and Asia (e.g., the Murray-Darling River, the Nile River, the Yellow River). Only three rivers (the Huai River and the Himalayan River in Asia, and the Po Valley in Europe) were identified to have innovation-driven structures. Water resources knowledge development in these rivers were considered diverse, with high focus on regional specific problems.

During the 1994 – 2005 period, the number of river basins with under-developed structures reduced to 27, mostly located in Asia. There were also more rivers identified as legacy-driven structures (40), covering most of the major river basins with high numbers of publications (80% of total publications, including the Yangtze River, the Mississippi River, the Great Lakes, the Nile River, and the Mediterranean Sea). Continued development of water resources knowledge was also evidenced as more rivers emerged with lower publications, but potentially demonstrated higher innovations (28), which were mainly located in Asia and North America (e.g., Indian River, Mackenzie River).

The knowledge structure of the water resources discipline has been highly developed during the most recent 2006 – 2017 period, with only 2 river basins were still considered to have under-developed structures, all of which located in North America (the Peace River, the James River). The number of river basins with legacy-driven structures (36) reduced during this period, most of which located in Asia or North America and still cover the major published river basins (80% of total publications) including the Yangtze River, the Mississippi River, the Great Lakes, the Pearl River, and the Yellow River. The river basins receiving top 5 publications in all time were include in this group, which implies the risk of centralised development of knowledge in highly researched river basins. On the other hand, multiple spatial centres were identified for increasing number of river basins with innovation-driven structures (57), spanning a broad spatial range in Africa (e.g., the Congo River), Asia (e.g., the Himalayan River), North America (e.g., the Yukon River), Europe (e.g., the Rhone River) as well as the Arctic and Antarctic Lakes. These rivers presented a higher tendency to spark radical innovations that addressed emerging management issues; yet there were also increasing risks of marginalisation of the core Water resources discipline in these river basins. It should be noted that no river basin was identified to have an ideal water resources knowledge structure during our whole study period.

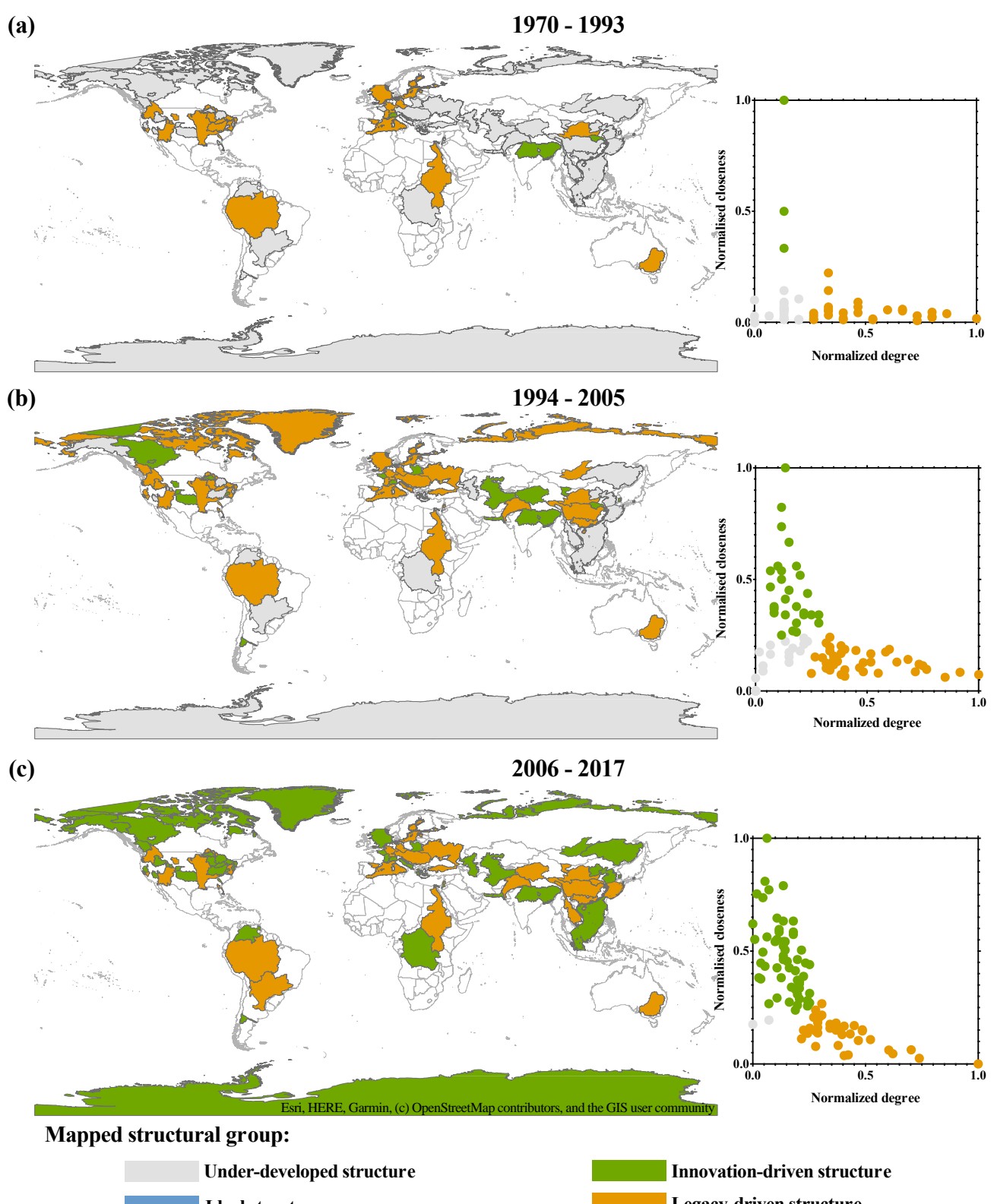

**(a)** 1970 - 1993

**(b)** 1994 - 2005

**(c)** 2006 - 2017

Esri, HERE, Garmin, (c) OpenStreetMap contributors, and the GIS user community

**Mapped structural group:**

| | | | |
|---|---|---|---|
| ⬜ Under-developed structure | | 🟩 Innovation-driven structure | |
| 🟦 Ideal structure | | 🟧 Legacy-driven structure | |

**Figure 4 The spatial distributions of knowledge structure of 95 river basins (left) and corresponding structural metrics (right) during (a) 1970 – 1993; (b) 1994 – 2005; (c) 2006 – 2017.**

**3.3 Relationship between researched management issues and structural development of the water resources discipline**

Almost 70% of river basins with under-developed structures indicated no clear management issue: i.e., the "Others" issue
group dominated during the 1970—1993 (Figure 5). Water pollution was the most prominent issue for the remaining under-developed river basins. The rivers with legacy-driven structures tended to focus on the issues related to surface water and groundwater management (e.g., the Colorado River), sedimentation and erosion (e.g., the Yellow River and the Ganga River)

and water pollution and treatment (e.g., the North Sea) as secondary issues of focuses. Only 3% of river basins were identified to have innovation-driven structures during this period, most of which focused on the water pollution issue.

During 1994-2005, as the number of river basins with under-developed structures reduced, river basins with the legacy-driven and innovation-driven structures demonstrated similar issues of focus: water pollution and treatment, and surface water and groundwater management (e.g., the Mackenzie River, the Arctic Lakes, the Jordan River). Moreover, interests on agricultural irrigation, droughts and floods, ecological degradation and restoration, and water policy had newly emerged for the innovation-driven structured river basins, covering the newly appeared key words in this period: "sediment", "nitrate", "water framework directive", "flow regulation", and "stakeholder management". For the legacy-driven structured river basins only the issue of the ecological degradation was newly considered (e.g., "ecological rehabilitation", "restoration"). Both types of river basins received limited studies on the climate variability and change, and droughts and floods issue.

During the most recent 2006 – 2017 period, both the legacy-driven and innovation-driven structured river basins reinforced their research interests on the surface water and groundwater management issue, represented by newly appeared key words including "SWAT", and "hydrodynamic model". As the research focuses of the legacy-driven structured river basins remained largely unchanged from the previous period, more innovation-driven river basin studies (e.g., in the San-Francisco Bay, the Haihe River) were conducted on newly emerged water pollution and treatment (e.g., "heavy metal", "saltwater intrusion"), the sedimentation and erosion (e.g., "sinkhole", "land loss"), climate variability and change (e.g., "global warming"), ecological degradation and restoration (e.g., "ecosystem health", "deforestation", "food web"), and new technologies developed (e.g., "remote sensing", "agent-based model") grouped under the others issue.

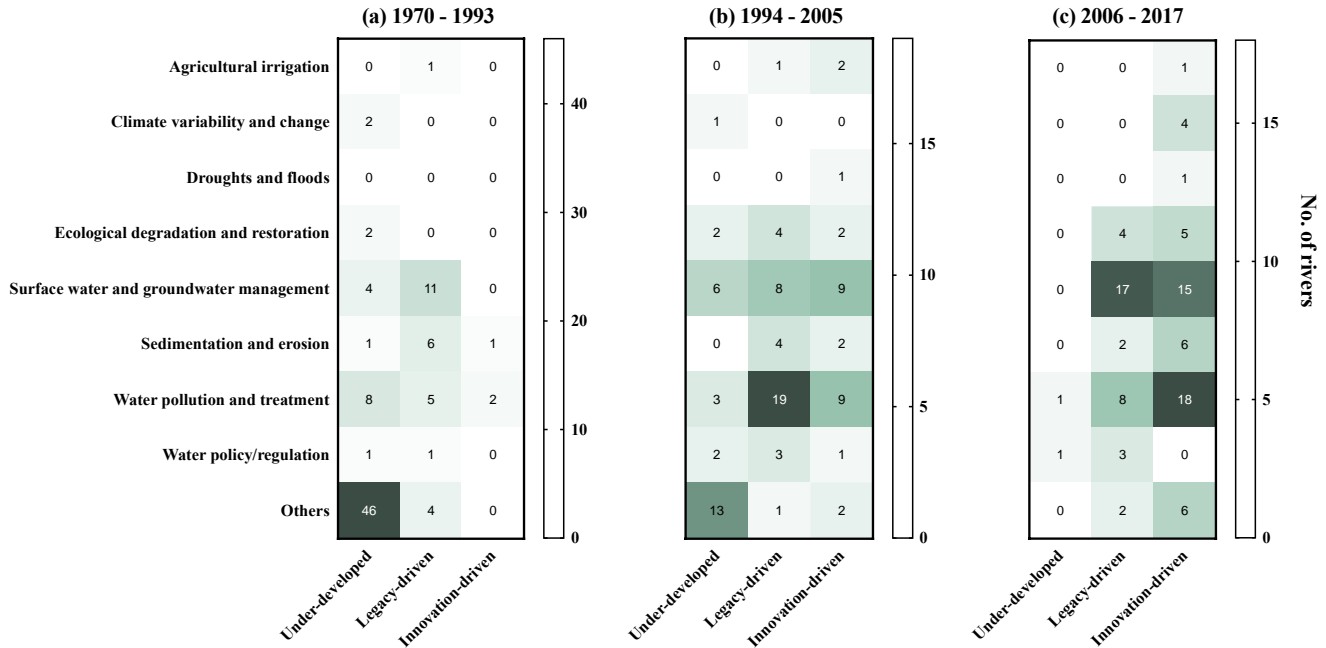

Note: "Others" group contains key words on pecific water bodies and/or technological terms which could not be grouped in any of the other issues above.

**Figure 5 Mapping of the river basins situated in the knowledge spectrum to their issues of focus during (a) 1970 – 1993; (b) 1994 – 2005; and (c) 2006 – 2017.**

**3.4 Cross-disciplinary collaborations of the water resources discipline**

Collaborations of the water resources discipline with other disciplines remained highly stable in time (Figure 6a). Environmental Science remained as the top one which the water resources discipline collaborated with in all 3 periods although the percentage in total publications reduced from 23% to 19%. It belonged to the category of life science and biomedicine, which also comprised over 50% of all collaborations of water resources during 1970 – 1993 (e.g., Environmental Sciences, Marine and Freshwater Biology, Ecology). Increasing cross-disciplinary collaborations of the water resources discipline were

identified with the physical sciences (over 30%, e.g., Multidisciplinary Geoscience, Oceanography, Meteorology and Atmospheric Sciences) since 1994-2005, and with engineering & technology (14%, e.g., Civil Engineering, Environmental Engineering, Remote Sensing) since 2006 – 2017. There was a gradual shift of disciplinary collaborations from biological and chemical-related disciplines to geographical and atmospheric-related. However, the proportions of collaboration with social sciences and arts and humanities remained at about 1% in all time, in another word, nearly no collaboration.

Matching the top 10 most published cross-disciplinary collaborations and the corresponding management issues, there identified high reliance of water resource knowledge development to collaborate with life sciences and biomedicines to solve all issues of focus, regardless the evolutions of the natural systems in time (Figure 6b). Environmental Sciences had been most relied on addressing the surface water and groundwater, sedimentation and erosion, water quality and treatment, and water policy issues, whereas Marine and Freshwater Biology was most connected to the ecological degradation and restoration issue.

Knowledge from Multidisciplinary Geosciences did not gain many publications regarding the surface water and groundwater issue until 2006 – 2017; while collaborations of the water resources discipline with Ecology and Multidisciplinary Geosciences have been sustained to solve agricultural irrigation, climate variability and change, and droughts and floods issues in all time. The dominance of life sciences and biomedicine in river basin studies was also evident spatially (Figure 6c). These disciplines contributed to between 40% to over 70% of global river basin studies, mostly for South American rivers (76%) and least for

Asian rivers (45%). Physical sciences contributed the most to Asian river studies (43%), whereas the proportions of contributions ranged between 20% and 40% for rivers in other continents. Technology and engineering disciplines were also mostly studied in Asian rivers (11%), followed by the North American rivers (8%), Oceania (Australia) (7%) and Europe (6%).

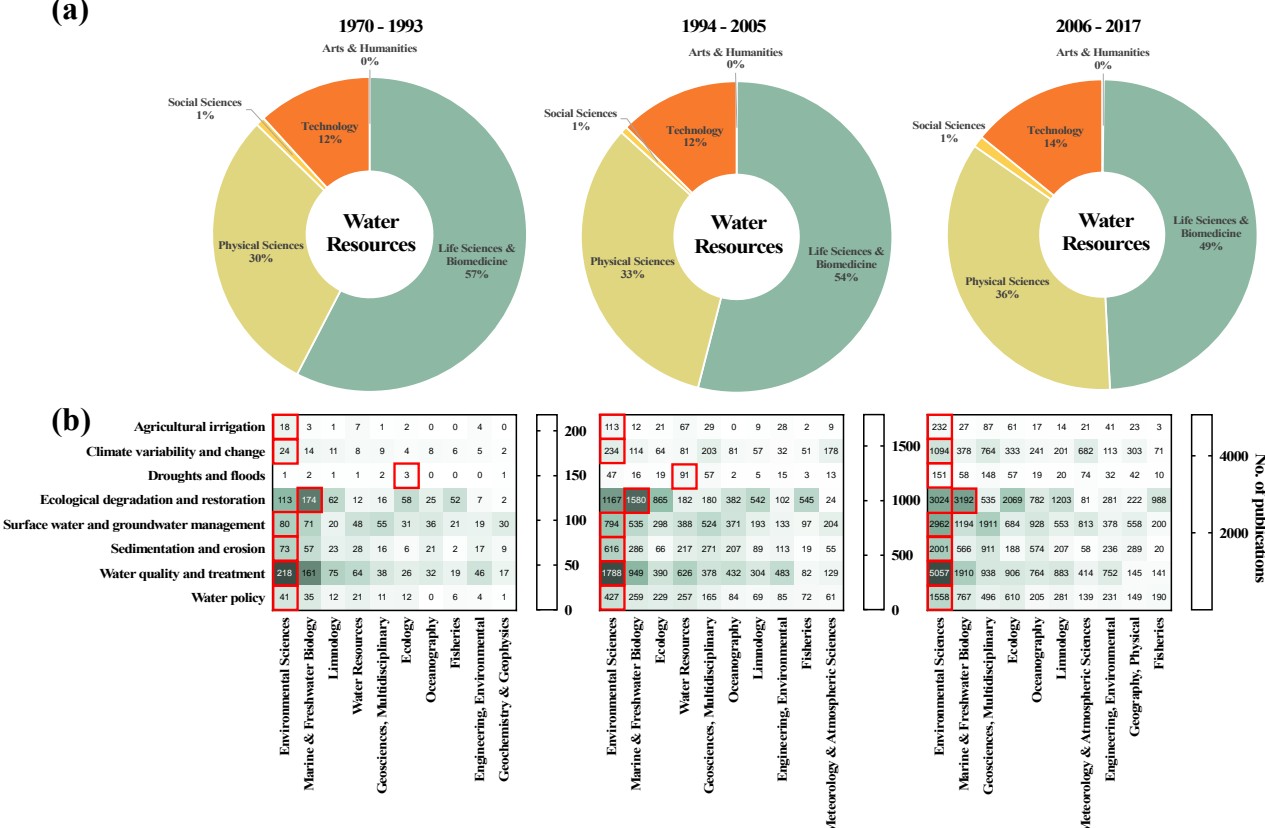

Note: "Other" group contains key words on specific water bodies and/or technological terms which could not be grouped in any of the other issues above, and hence not included for presentation.

Only the top 10 most published disciplines in each temporal period are shown. Red square indicates highest publication for each issue and the corresponding discipline.

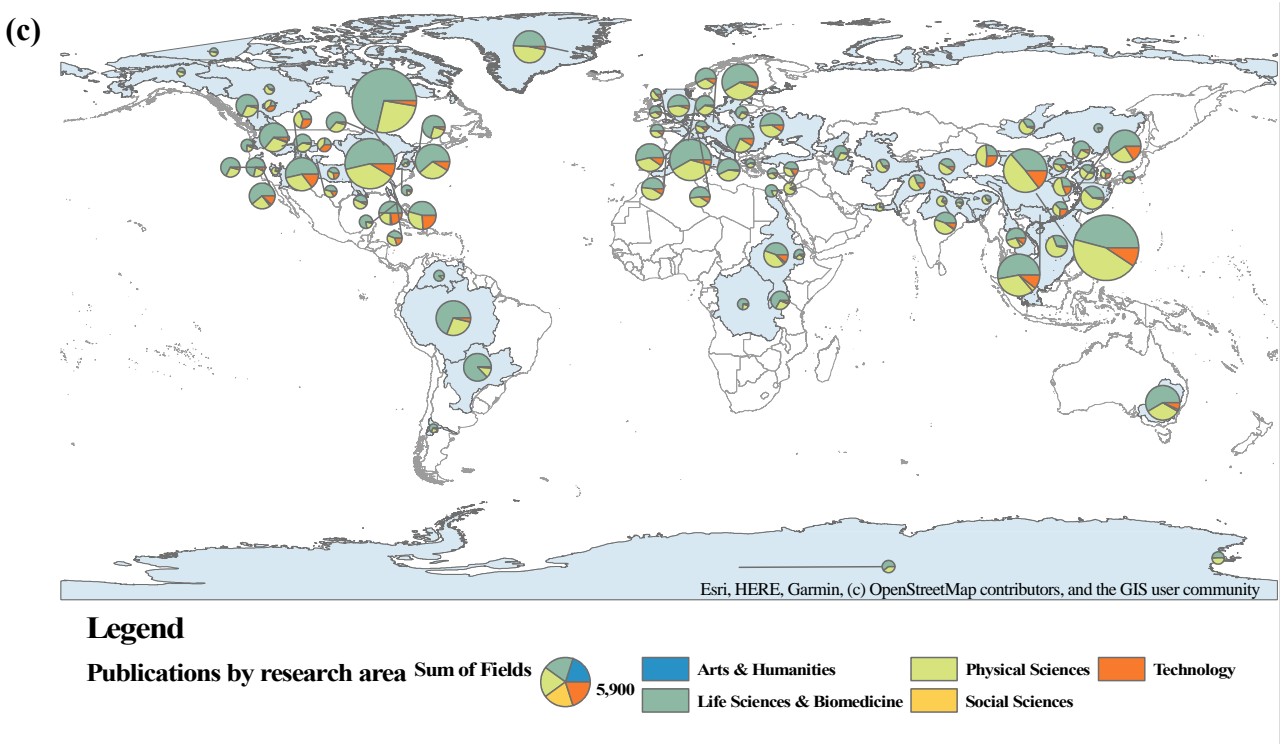

**Figure 6 (a) The evolution of cross-disciplinary collaborations with the water resources discipline; (b) the evolution of links between management issues and the collaborations of the water resources discipline during 1970 – 1993, 1994 – 2005, and 2006 – 2017; and (c) the spatial distributions of disciplines by research area in all time.**

**4. Discussion**

Using the academic publications indexed in the WoS database between 1900 – 2017 as the data source, this study investigated the development of water resources knowledge at the global river basins scale from the structure perspective. Key findings are summarised below:

- Three development periods were identified for the water resources knowledge system: 1900-1993, 1994-2005, and 2006-2017. Studies on major rivers in Europe and North America dominated the early development, while those on Asian rivers were catching up quickly since 2000s.

- The water resources knowledge system was highly skewed towards a legacy-driven structure, dominated by the top 10 river basins that accounted for 48% of total publications in the 2006-2017 period (e.g., the Yellow River, the Yangtze River, the Mississippi River, the Murray-Darling River, and the Mediterranean Sea). On the other hand, 57 river basins demonstrated innovation-driven structures with focus on emerging issues (e.g., climate change). While less published, these rivers were globally distributed. No river basin was identified to have an ideal water resources knowledge structure during the study period.

- The management issues were increasingly homogenised particularly for the river basins with legacy-driven knowledge structures.

- Collaborations of the water resources discipline with other disciplines overwhelmingly dominated by Environmental Sciences, Marine and Freshwater biology, Ecology, Multidisciplinary Geosciences, and Environmental Engineering, whereas collaborations with social sciences remained very limited in all time.

During recent decades, great advances have been made on the water resources discipline. It has evolved from an engineering focus on empirical estimation of water and its flow processes in the 1970s (Sivapalan, 2018); integrating with ecology and meteorology with a focus on changes in vegetation and habitats under the impacts of climate and land use changes in the 1990s (Rodriguez-Iturbe, 2000); to the most current approach to understand the social and economic system impacts on the water cycle after 2000s (Montanari et al., 2015). These endeavours have led to prosperity of a wide range of sub-disciplines including the eco-hydrology, hydro-meteorology, and socio-hydrology, utilising advanced observational and computational technologies such as remote sensing, global-scale hydrological modelling, agent-based modelling and convolutional neural network modelling (Mccurley and Jawitz, 2017; Xu et al., 2018; Savenije et al., 2014). Complementing these knowledge development, our findings are of several implications from a knowledge system structural perspective. Specifically, 1) the finding that water resources knowledge development was dominated by the legacy-driven knowledge structure indicates that the current knowledge system was strongly supported by existing theories and methods and that the diversity as an important feature of a good system structure is missing (Biggs et al., 2015; Park et al., 2021). This structure could have strong diffusive power, but there is risk of knowledge redundancy that could hinder innovation and potential waste of research resources (Makri et al., 2010). 2) The tendency of legacy-driven river basins focusing on the same management issues further increases the risk of homogenization and reduces the resilience (capacity) of the water resources knowledge system to address problems arising from the abruptly changing environment. 3) Collaborations of the water resources discipline with social sciences were very limited further indicate that the existing knowledge system did not support knowledge inputs from diverse disciplines. For example, the socio-hydrology emerged as a new sub-discipline of water resources in 2012 to understand the coupled human-water relationships by integrating knowledge from social sciences into hydrology (Sivapalan et al., 2012; Di Baldassarre et al., 2013), and was further advocated by the IAHS's most recent scientific decade: "Panta Rhei 2013-2022" to understand the co-evolutions of social, cultural, economic, political and physical dimensions of water (Mcmillan et al., 2016; Savenije et al., 2014; Di Baldassarre et al., 2019). However, it was found from our structural analysis that during the 2006-2017 period, the water resources discipline was only linked to a limited number of social science disciplines with the most prominent being Geography (Human), Economics, Planning & Development, accounting for only 1% of total collaborations but over 50% of

social sciences collaborations. There was none to very limited links with Psychology, Behavioural Science, Sociology, Art, Cultural Studies, Political Science, International Relations, Law and Public Administration, which are core disciplines for explaining individual and collective human behaviours. This implies that the existing knowledge structure did not fully support the development of socio-hydrology. 4) More than half of river basins studied (57) presented innovation-driven structures in the latter period. These river basins offer an opportunity to generate innovations in the water resources knowledge system by strengthening those weak links with social sciences and building on their existing studies on diverse and regional-specific issues, although the marginalization of the core disciplines should be avoided. 5) No river basin was identified to have an ideal knowledge structure (i.e. appropriate centrality and diversity) during the study period. It implies that the existing water resources knowledge structure could not support both innovation and legacy to a high level. All of these findings indicate that the knowledge structure should be taken into account in the strategic design and planning of future research on the water resources discipline.

The current water resources knowledge structure is an accumulated product of intrinsic factors and extrinsic drivers. It is widely recognized that the knowledge development is intrinsically influenced by the philosophy, ontology, epistemology of research communities (Kuhn, 1996; Ludwig and El-Hani, 2020). Philosophically, a few hydrologists (e.g., Sivapalan and Blöschl, 2017) have argued that our water resources knowledge system should enter punctuated growth in its evolutionary cycles of punctuated equilibria (Gould and Eldredge, 1972), and its euphoria should close to be ended with the disenchantment as current knowledge is not sufficient to address the emerging global challenges. Meanwhile, a majority of hydrologists insist that the fundamental unsolved scientific questions in water resources system remain the same (Blöschl et al., 2019). It seems that radical departures from the past path is not likely in the near future due to lack of intrinsic push. Ontologically (regarding the conception of reality), although there have been increasing interests to integrate sociology, law, history, psychology and other social sciences into the water resources discipline, different scientific communities have different ontological perceptions regarding river management issues (Castillo et al., 2020). Epistemologically (regarding the conception of science), there is a general belief that natural sciences strive for quantitative generalizations and modelling of the biophysical processes, while social sciences tend to focus on qualitative case studies to understand the contexts of human interventions (Ayllón et al., 2018; Malek and Verburg, 2020). These are huge challenges for transforming the water resources knowledge structure.

Challenges also come from extrinsic drivers. Academic capitalism (market driven and market-like activities), which have been highly skewed towards natural sciences, is one of the most direct causes (Nickolai et al., 2012; Slaughter and Rhoades, 2004). The metrics-driven evaluation of scientific activities widely implemented by research institutes and universities is another cause (Louder et al., 2021; Muller, 2018). Overemphasis on simple evaluations provide incentives to tailor research to meet the metrics, and social science publications are being more marginalized as their journals tend to have much lower impact factors. In addition, regardless the constant calls on interdisciplinary research in the past decades (e.g., Gleick, 2000; Caldas et al., 2015), the funding ratio to support social sciences (about 30%) are significantly lower than that for the natural sciences (over 80%) in most countries (Xu et al., 2015). More importantly, as argued by this study, the knowledge structure has not been given enough importance in the strategic design and planning of research priorities.

Finally, limitations of this study should be noted. Firstly, by choosing river basin as the unit of study, it is possible that publications on general conceptual/theoretical development without specific spatial links and those publications at global scale may be missed; and by limiting the study scope to the 95 most researched river basins indexed in the WoS database in English may also narrow the coverage of this study on water resources knowledge development. Furthermore, not including non-academic documents (e.g., government reports or consultation reports) may miss the practice-driven knowledge developed in river basin studies. Secondly, we recognise the blurring of disciplinary boundaries particular for cross-disciplinary journals in the classifications of the WoS database. Finally, the types of knowledge structures were empirically determined by the AHC algorithm in this study, more theoretical support should be sought, which is our future research direction.

**5. Conclusion**

To conclude, the stationarity of the hydrological systems is dead, but the stationarity of the water resources knowledge structure persists. This knowledge system is dominated by highly researched river basins with legacy-driven knowledge structures, homogenized structure-issue links, and stabilized disciplinary collaborations with limited contributions from social sciences. A structural shift of water resources knowledge towards social sciences is required to support sustainable river basin development in the Anthropocene.

**Appendix A**

The five subject categories and corresponding disciplines classified under each category as outlined in (Clarivate Analytics, 2018) are summarised in Table A.1.

Table A.1 Classification of subject categories and disciplines

| Category | Discipline |
| --- | --- |
| Arts & Humanities | Architecture; Art; Arts & Humanities Other Topics; Asian Studies; Classics; Dance; Film, Radio & Television; History; History & Philosophy of Science; Literature; Music; Philosophy; Religion; Theatre. |
| Life Sciences & Biomedicine | Agriculture; Allergy; Anatomy & Morphology; Anaesthesiology; Anthropology; Audiology & Speech-Language Pathology; Behavioural Sciences; Biochemistry & Molecular Biology; Biodiversity & Conservation; Biophysics; Biotechnology & Applied Microbiology; Cardiovascular System & Cardiology; Cell Biology; Critical Care Medicine; Dentistry, Oral Surgery & Medicine; Dermatology; Developmental Biology; Emergency Medicine; Endocrinology & Metabolism; Entomology; Environmental Sciences & Ecology; Evolutionary Biology; Fisheries; Food Science & Technology; Forestry; Gastroenterology & Hepatology; General & Internal Medicine; Genetics & Heredity; Geriatrics & Gerontology; Health Care Sciences & Services; Hematology; Immunology; Infectious Diseases; Integrative & Complementary Medicine; Legal Medicine; Life Sciences Biomedicine Other Topics; Marine & Freshwater Biology; Mathematical & Computational Biology; Medical Ethics; Medical Informatics; Medical Laboratory Technology; Microbiology; Mycology; Neurosciences & Neurology; Nursing; Nutrition & Dietetics; Obstetrics & Gynaecology; Oncology; Ophthalmology; Orthopaedics; Otorhinolaryngology; Palaeontology; Parasitology; Pathology; Paediatrics; Pharmacology & Pharmacy; Physiology; Plant Sciences; Psychiatry; Public, Environmental & Occupational Health; Radiology, Nuclear Medicine & Medical Imaging; Rehabilitation; Reproductive Biology; Research & Experimental Medicine; Respiratory System; Rheumatology; Sport Sciences; Substance Abuse; Surgery; Toxicology; Transplantation; Tropical Medicine; Urology & Nephrology; Veterinary Sciences; Virology; Zoology. |
| Physical Sciences | Astronomy & Astrophysics; Chemistry; Crystallography; Electrochemistry; Geochemistry & Geophysics; Geology; Mathematics; Meteorology & Atmospheric Sciences; Multidisciplinary Geosciences; Mineralogy; Mining & Mineral Processing; Oceanography; Optics; Physical Geography; Physics; Polymer Science; Thermodynamics; Water Resources. |

| Management issue | Example key words |
|---|---|
| Social Sciences | Archaeology; Area Studies; Biomedical Social Sciences; Business & Economics; Communication; Criminology & Penology; Cultural Studies; Demography; Development Studies; Education & Educational Research; Ethnic Studies; Family Studies; Geography (Human); Government & Law; International Relations; Linguistics; Mathematical Methods In Social Sciences; Psychology; Public Administration; Social Issues; Social Sciences Other Topics; Social Work; Sociology; Urban Studies; Women's Studies. |
| Technology (Engineering) | Acoustics; Automation & Control Systems; Computer Science; Construction & Building Technology; Energy & Fuels; Engineering; Imaging Science & Photographic Technology; Information Science & Library Science; Instruments & Instrumentation; Materials Science; Mechanics; Metallurgy & Metallurgical Engineering; Microscopy; Nuclear Science & Technology; Operations Research & Management Science; Remote Sensing; Robotics; Science & Technology Other Topics; Spectroscopy; Telecommunications; Transportation. |

## Appendix B

Each management issue and corresponding example key words included in it are summarised in Table B.1.

Table B.1 Summary of identified management issues and example key words in each issue group

| Management issue | Example key words |
|---|---|
| **Agricultural irrigation**: Related to specific irrigation and farming methods and techniques, including agriculture, horticulture, and animal husbandry. | "conservation tillage", "grain yield", "grass seed production", "paddy field", "energy crop", "milk quality", "silviculture", "global production networks" |
| **Climate variability and change**: Related to climatic, atmospheric, and meteorological changes. | "arid region", "climate change", "climate warming", "cold fronts", "global warming", "heat flow", "El Niño", "atmospheric circulation", "tropicalization", "adaptive radiation" |
| **Droughts and floods**: Refers to specific mentions of floods and droughts. | "flood pulse", "flood risk", "drought", "paleo flood" |
| **Ecological degradation and restoration:** Related to the ecosystem and their restorations. | "bioavailability", "biodegradation", "ecological risk", "ecosystem health", "food web", "forest value chain", "deforestation", "harmful algal bloom", "spawning migration" |
| **Erosion and sedimentation**: Related to the processes and changes in earth surface. | "sedimentation", "bank erosion", "bottom sediments", "deposition", "erosion", "fluvial process", "resuspension", "soil erosion", "suspended particulate matter", "sediment" |
| **Surface water and groundwater management**: Related to hydrological processes and changes in water resources in both surface water and groundwater. | "water level fluctuations", "drainage", "groundwater depletion", "sewer overflows", "backwaters", "flow regime", "evapotranspiration", "river discharge", "river-lake interaction", "precipitation" |
| **Water policy/regulation** | "water resources planning", "integrated management", |

| | |
|---|---|
| Refers to water policy initiatives, governance, and broad human activities related to water | "human activity", "hydropolitics", "hydropower development", "demand management", "stakeholder management", "water framework directive" |
| **Water pollution and treatment**:<br>Refers to pollutions and corresponding treatments. | "mercury", "acid", "pollution", "contaminated loads", "chlorinated compound", "Cyanobacterial blooms", "heavy metal", "high-level waste", "methane emission", "denitrification" |
| **Others (not elsewhere classified)**:<br>Refers to specific water bodies and/or technological terms that cannot be classified in any other issues. | "comparative study", "agent-based modelling", "cross-sectional study", "integrated case study", "remote sensing", "Dead Sea coast", "Ganga river system", "Mississippi river plume", "Yellow river water", "Ohio river water" |

**Appendix C**

For any node d (a specific discipline) in the network:

Degree = Sum of no. adjacent edges connected to d;                                                      (C.1)

Closeness = 1/ Sum of the shortest path of d to/from all other nodes (i)

$\qquad$ = 1 / $\Sigma$ shortest distance between (d,i), where i ≠ d;                                   (C.2)

To facilitate comparability of the two measures among the 95 river basins, the values of degree and closeness for the water

resources discipline were normalised using Eq. C.3:

$$Normalised\ k_i = \frac{(raw\ k_i - min.\ k)}{(max.\ k - min.\ k)}$$                                   (C.3)

**Data and code availability**

Data and codes generated in this study can be accessed via: https://doi.org/10.7910/DVN/GWXWMB.

**Author contribution**

S. Wu contributed to conceptualization, methodology development, data analysis, writing of original the draft and reviewing and editing of the manuscript. Y. Wei contributed to conceptualization, methodology development, data analysis, reviewing and editing of the manuscript, and funding acquisition. X. Wang contributed to data curation and validation, and reviewing of the manuscript.

**Competing interests**

The authors declare that they have no conflict of interest.

**Acknowledgement**

This study is supported by Australian Research Council Special Research Initiative for Australian Society, History and Culture [SR200200186].

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
