# Peer review of "Global water resources knowledge gaps"

_Hydrology and Earth System Sciences, 2021_

## Community Comment (CC1)

Dear Professor Savenije,

Thank you very much for your insightful comments on our manuscript. We would like to take this opportunity to address your comments by clarifying those points that were not clear in our manuscript and proposing how we will improve our manuscript based on your comments.

*I do understand that this is a data-mining exercise, and that the authors did not necessarily familiarize themselves with the field of Water Resources Management and its development over time. I recommend looking at the paper on "Evolving water science in the anthropocene" (https://hess.copernicus.org/articles/18/319/2014/) and the huge body of papers that have recently been published under the IAHS research initiative "Panta Rhei", e.g. "Global perspectives on hydrology, society and change", Hydrological Sciences Journal, 61:7, 1174-1191, (DOI:10.1080/02626667.2016.1159308).*

Thank you very much for your comments. We will revise our introduction section by making our motivation and approach clearer as briefly discussed below.

**Our motivation**

Humans have made substantial impacts on various Earth system cycles, marking the transition of our planet into the Anthropocene (Crutzen, 2002; Crutzen & Stoermer, 2000). This has been powered by developments of science and technology in particular since the Industrial and Scientific Revolutions (Lewis & Maslin, 2015; Lubell & Morrison, 2021; Steffen et al., 2011). Rethinking scientific development in the Anthropocene is crucial for our future survival. Hydrological cycles are a central component of the Earth system and it is widely recognized that the stationarity of hydrological systems is dead as a result of human activities (Ajami et al., 2017; Birkinshaw et al., 2014; Milly et al., 2008). Therefore, to **investigate the knowledge gap of hydrology/water resources from its evolutionary history could increase our capacities adaptive to transition into the Anthropocene**.

**Our approach**

The hydrology/water resources knowledge is a complex disciplinary system and a sub-system of the entire knowledge system covering all scientific disciplines. It is recognised that the functionality of a complex system depends on its structure (Huttenhower et al., 2012; Sayles & Baggio, 2017; Von Bertalanffy, 1968). The disciplinary knowledge structure is often analysed in two primary ways. First, discipline experts qualitatively review and assess theoretical advances, technology (methods and instruments) development and key challenges in the field based on their research experiences and professional knowledge (e.g. McMillan et al., 2016; Savenije et al., 2014; Sivapalan, 2018). Second, systematic bibliometric studies are conducted to quantitatively survey the structure of disciplinary knowledge and reveal the interactions among major research topics (e.g. Zare et al., 2017; Zeng et al., 2017). This paper aims to investigate the hydrology/water resources knowledge structure development using the complex network system approach on bibliometric data **to complement to those findings from existing professional knowledge and research experience and identify the potential gaps of knowledge structures of major river basins in the world.**

Specifically, we will examine the development of hydrology/water resources knowledge structure on:

- Evolution of management issues (study objects) by temporal stages;
- Evolution of the disciplinary structure by temporal stages;
- Links between the disciplinary structure and the management issues; and
- Collaborations of hydrology/water resources disciplines with other disciplines.

In addition, we will also use these recommended references along with others to discuss the implications of our findings from the perspective of complement to existing findings of professional knowledge and research experience in a later paragraph related to the Discussion section.

*The authors have analysed papers categorised in the WoS under Water Resources and limited the analysis to articles that deal with river basins or catchments in a broad sense. They then looked for connections between disciplinary fields of WRM and analysed the connections between these fields and*

*how they developed over time. They then classified the patterns of interconnection, or lack thereof into knowledge structures with the following names: Isolated, Innovative-inclined, Legacy-inclined and Centralised. To me, these classifications have hardly any explanatory power. I have gone through the description several times, but I fail to see what these terms actually mean or imply in relation to WRM. I can't see whether they have a positive or negative connotation. To me, Isolated and Centralised sounds rather negative; Innovative-inclined sounds positive; and Legacy-inclined may be both positive or negative, depending on one's perspective. That in recent years more basin studies are legacy-inclined may be evidenced by the data, but I have difficulty to see what it means.*

Thank you for your comments. We will revise our data and methods section to further clarify the knowledge structure metrics and their significances as briefly discussed below.

To clarify the significance of different knowledge structures to river basin management practices, **we will reorganise our classification of knowledge structure into four types using the two commonly used metrics in the system network theory: centrality and diversity.** Centrality measures the number of connection a node has in a knowledge network system, reflecting the level of knowledge concentration: the greater the centrality, the more connected a discipline is and thus more concentrated. Diversity measures the inverse sum of connecting distances to all other nodes, reflecting the extent to which a node is isolated within the knowledge system: the greater the diversity, the fewer extended connections a discipline has and thus forming more confined small groups in the network. Empirical analyses have demonstrated that concentrated knowledge structures facilitate dissemination of existing knowledge, whereas isolated structure can increase adaptivity to different disciplinary knowledge and facilitate radical innovations to knowledge development through looking from divergent angles (Bodin & Prell, 2011; Foray, 2018; Schot & Geels, 2008).

Based on the differences between the centrality and diversity values, four types of knowledge structures can be identified (Figure 1). They are:

1. Ideal structure with high centrality and high diversity. With this structure, the river basin should have high research intensities in core disciplines to provide solid theoretical foundations, while at the same time sufficient cross-disciplinary collaborations to ensure knowledge innovations to address unexpected, emerging river basin management challenges.

2. Innovation-inclined structure with high diversity but low centrality, which could have a risk of discipline hollowing-out (marginalization of influence of core discipline). For the river basins with this structure, the connection with core disciplines (centrality) should be strengthened.

3. Legacy-driven structure with high centrality and low diversity, which discourages knowledge innovation. In the river basin with this structure, the cross-disciplinary collaborations (diversity) should be strengthened to increase the potential of knowledge pattern transformation against emerging management challenges; and

4. Underdeveloped structure with low centrality and low diversity, indicating that the knowledge development is still at its early stage and the knowledge development should be strengthened comprehensively.

[Figure]

**Figure 1** Four different knowledge structures based on their structural metrics

Therefore, **classifying the knowledge structure of the river basins into four different knowledge structural types enables direction of the strategic design and planning of future research from the structural perspective.**

*By choosing to analyse traditional disciplinary fields, such as: Agricultural irrigation; Erosion and sedimentation; Water pollution and treatment; Surface water and groundwater management; Ecological degradation; Droughts and floods; Climate variability and change; the results obtained are hardly pointing towards stronger societal linkages. I miss emerging new fields, such as: demand management; decentralisation; participation; international water law; .... and new technologies such as Remote Sensing, New observation technology, Global modelling, Artificial intelligence, .... If you look for traditional terms, you are bound to find traditional results.*

Thank you for your comments. To clarify, the nine topics we identified were derived from our data rather than pre-set. To comprehensively reflect the management issues in our methods section, we have extracted key words from the sections of Title, Abstract and Keywords rather than only the Keywords section of each publication using text-mining techniques. These key words were extracted if they have high weighting values on their Term Frequency-Inverse Document Frequency (TF-IDF). TF-IDF was calculated to give higher weights to key words with a high appearance frequency in its corresponding section and a low overall appearance frequency in the entire text collection to avoid a bias towards general terms and grasp the newly appeared key words.

In addition, **we will re-examine and highlight those newly appeared key words in each temporal stage and may categorise them in newly defined groups to more precisely reflect the evolution of management issues.**

*By taking river basins as the entree point, I fear the authors have missed a huge body of conceptual and global research. Not all WR research is done at river basin level. Much happens at the global scale, national scale, policy scale or conceptually.*

We chose river basin as the spatial unit for analysis as it represents the territorial unit of water cycle linking to other cycles of the Earth system (e.g. nutrients, energy, and carbon), which are commonly adopted by researchers to understand the integrated impacts of water use, land use and environmental management (Newson, 2008; Warner et al., 2008). We merged those publications focusing smaller spatial units (e. g. sub-catchment, or wetland or lake into the river basin which they are affiliated with). **But we agree that we may have missed the publications on general conceptual/theoretical development without specific spatial links and those publications at global scale. Thus, we will revise our manuscript title as "Gaps of knowledge structure in river basins".**

Part of results will be revised based on the change of methods as discussed above.

*Section 4. Discussion and Conclusion, is hardly a discussion. It is rather a set of three recommendations where certain lines of research are "encouraged": 1) investigation of "new" river basin phenomena; 2) spatial diversity of Water Resources research; and 3) strengthening collaborations with social sciences. These are rather obvious and general recommendations and hardly a discussion. The conclusion that "the stationarity of the water resources knowledge system persist" is not supported by the large body of work that is recently being produced as a result of and as part of the "Panta Rhei" initiative. This large body of work is hard to detect if one constrains oneself to the 95 most studied river basins in the world and the connections to traditional fields.*

Thank you for your comments. We will revise our three recommendations based on the results to be updated and draw the conclusion from **the perspective of knowledge structure** of river basin.

We will discuss the implications of the key findings from the following aspects:

1. **We will discuss the implications of our findings from the perspective of complement to existing findings of professional knowledge and research experience.** This will be done by briefly summarizing the evolution of management issues and discipline of hydrology/water resources knowledge structure by reviewing the key findings from the IAHS Scientific Decades since its foundations in particular the most current "Panta Rhei" Decade, covering the studies you recommended and major review articles published in the Water Resource Research, HESS and other hydrological journals.

2. **We will discuss the intrinsic cause of current knowledge structure by briefly discussing the challenges in integrating hydrology with social sciences from the epistemological, ontological, and political perspectives,.**

3. **We will discuss the extrinsic cause of current knowledge structure from current academic capitalism; and**

4. **Finally, we will discuss the limitations of our study including only journal papers were considered and focusing on river basin scale studies.**

Once again, thank you very much for your valuable comments.

Your sincerely,

The authors team:
Shuanglei Wu; Yongping Wei; Xuemei Wang.

**References**

Ajami, H., Sharma, A., Band, L. E., Evans, J. P., Tuteja, N. K., Amirthanathan, G. E., & Bari, M. A. 2017. On the non-stationarity of hydrological response in anthropogenically unaffected catchments: an Australian perspective. *Hydrol. Earth Syst. Sci.,* 21(1), 281-294. doi:10.5194/hess-21-281-2017

Birkinshaw, S. J., Bathurst, J. C., & Robinson, M. 2014. 45 years of non-stationary hydrology over a forest plantation growth cycle, Coalburn catchment, Northern England. *Journal of Hydrology,* 519, 559-573. doi: 10.1016/j.jhydrol.2014.07.050

Bodin, Ö., & Prell, C. 2011. *Social Networks and Natural Resource Management : Uncovering the Social Fabric of Environmental Governance*. Cambridge, UNITED KINGDOM: Cambridge University Press.

Crutzen, P. J. 2002. Geology of mankind. *Nature*, 415(6867), 23-23. doi:10.1038/415023a

Crutzen, P. J., & Stoermer, E. F. 2000. The Anthropocene. *IGBP Clobal Change News Letter*, 41, 17-18.

Foray, D. 2018. Smart specialization strategies as a case of mission-oriented policy—a case study on the emergence of new policy practices. *Industrial and Corporate Change*, 27(5), 817-832. doi: 10.1093/icc/dty030

Huttenhower, C., Gevers, D., Knight, R., Abubucker, S., Badger, J. H., Chinwalla, A. T., . . . The Human Microbiome Project, C. 2012. Structure, function and diversity of the healthy human microbiome. *Nature*, 486(7402), 207-214. doi:10.1038/nature11234

Lewis, S. L., & Maslin, M. A. 2015. Defining the Anthropocene. *Nature,* 519(7542), 171-180. doi:10.1038/nature14258

Lubell, M., & Morrison, T. H. 2021. Institutional navigation for polycentric sustainability governance. *Nature Sustainability*. doi:10.1038/s41893-021-00707-5

McMillan, H., Montanari, A., Cudennec, C., Savenije, H., Kreibich, H., Krueger, T., . . . Xia, J. 2016. Panta Rhei 2013–2015: global perspectives on hydrology, society and change. *Hydrological Sciences Journal*, 61(7), 1174-1191. doi:10.1080/02626667.2016.1159308

Milly, P. C. D., Betancourt, J., Falkenmark, M., Hirsch, R. M., Kundzewicz, Z. W., Lettenmaier, D. P., & Stouffer, R. J. 2008. Stationarity Is Dead: Whither Water Management? *Science*, 319(5863), 573-574. doi:10.1126/science.1151915

Newson, M. 2008. *Land, water and development: sustainable and adaptive management of rivers*: Routledge.

Savenije, H. H. G., Hoekstra, A. Y., & van der Zaag, P. 2014. Evolving water science in the Anthropocene. *Hydrol. Earth Syst. Sci.,* 18(1), 319-332. doi:10.5194/hess-18-319-2014

Sayles, J. S., & Baggio, J. A. 2017. Social–ecological network analysis of scale mismatches in estuary watershed restoration. *Proceedings of the National Academy of Sciences*, 114(10), E1776-E1785. doi:10.1073/pnas.1604405114

Schot, J., & Geels, F. W. 2008. Strategic niche management and sustainable innovation journeys: theory, findings, research agenda, and policy. *Technology Analysis & Strategic Management*, 20(5), 537-554. doi:10.1080/09537320802292651

Sivapalan, M. 2018. From engineering hydrology to Earth system science: milestones in the transformation of hydrologic science. *Hydrol. Earth Syst. Sci.*, 22(3), 1665-1693. doi:10.5194/hess-22-1665-2018

Steffen, W., Grinevald, J., Crutzen, P., & McNeill, J. 2011. The Anthropocene: conceptual and historical perspectives. *Philosophical Transactions of the Royal Society A: Mathematical, Physical and Engineering Sciences*, 369(1938), 842-867. doi: 10.1098/rsta.2010.0327

Von Bertalanffy, L. 1968. *General system theory*. New York Magazine, 41973(1968), 40.

Warner, J., Wester, P., & Bolding, A. 2008. Going with the flow: river basins as the natural units for water management? *Water Policy*, 10(S2), 121-138. doi:10.2166/wp.2008.210

Zare, F., Elsawah, S., Iwanaga, T., Jakeman, A. J., & Pierce, S. A. 2017. Integrated water assessment and modelling: A bibliometric analysis of trends in the water resource sector. *Journal of Hydrology*, 552, 765-778. doi: 10.1016/j.jhydrol.2017.07.031

Zeng, A., Shen, Z., Zhou, J., Wu, J., Fan, Y., Wang, Y., & Stanley, H. E. 2017. The science of science: from the perspective of complex systems. *Physics Reports*, 714-715, 1-73. doi: 10.1016/j.physrep.2017.10.001

---

## Community Comment (CC2)

**Dear Dr. Xu,**

Thank you very much for your insightful comments on our manuscript. We would like to take this opportunity to address your comments by clarifying the unclear points and proposing how we will improve our manuscript.

**General comments**

This work aimed to discover knowledge gaps in water resources research at the river basin scale through looking into the knowledge structure and disciplinary connections over time. The starting point of this paper is very interesting and the topic is important as river management and governance are highly fragmented. Generalizing knowledge patterns for research and management practices at the basin scale is challenging but should be done. Identification of knowledge gaps through investigating the knowledge structure is an innovative approach. Tracing the knowledge development patterns could also help identify gaps between science and policy, which is critical for the knowledge mobilization that promotes science-based decision-making for water systems. The synthesis of such fragmented knowledge would be benefited from large data analytics such as text mining approaches and content analysis. Text mining is an efficient way for the synthesis of knowledge which otherwise will be buried in the large number of texts.

This paper used academic literature obtained from the Web of Science as the main source and made use of a text-mining approach to extract key terms from the literature. The authors then used two indicators (degree and closeness) to measure connections among knowledge domains defined in this study. Overall, the methodology is designed in a reasonable manner and discussions are fair. However, some revisions are required to make it more readable and informative.

**Thank you very much for your positive comments.**

Knowledge structure is a keyword of the paper and it is a cognitive concept/science which needs to be carefully defined. It has been well defined in many other disciplines such as education, psychology, etc. What does it mean in water science at the basin scale?

Thank you for your comment. Knowledge is typically recognised as a system. Scientific knowledge represents "ordered knowledge of phenomena and the rational study of the relations between the concepts in which those phenomena are expressed" (Dampier, 1944). Recently, scientific knowledge is increasingly recognised as a complex and dynamic network in which scientists, disciplines and phenomena to be "weaved together into an overarching scientific fabric" (Coccia, 2020; Shi et al., 2015). The complex interdependencies in the fabric are considered as the structure of knowledge. In our context, we define the knowledge structure in a river basin as a co-evolutionary process involving scientific disciplines and management issues which have their respective evolutionary dynamics. We will include more detailed definition of knowledge structure in the introduction section.

It will be beneficial for the paper to list definitions of terms in a table (i.e., limited development, isolated development, innovative-inclined development, legacy-inclined development, centralised development). As these terms are not commonly used in the context of water sciences, nor is it in knowledge evolution, one might need to go back to read definitions a few times before he/she could understand and remember them. If they are new to the field, the authors should make them clearer to be understood. A diagram that distinguishes them from each other would be helpful as they are now ambiguous. Alternatively, the authors may need to rephrase them into terms that are more common (e.g., "lack of knowledge", "disciplinary", "multidisciplinary", "interdisciplinary", "transdisciplinary", etc. Tress et al., 2005. Clarifying integrative research concepts in landscape ecology. Landscape Ecology, 20, 247-493).

Thank you for your comments. As we stated in our response to Professor Savenije, we will reorganise our classification of knowledge structure into four types using the two commonly used metrics in the system network theory: centrality and diversity (Figure 1).

Centrality measures the number of connection a node has in a knowledge network system, reflecting the level of knowledge concentration: the greater the centrality, the more connected a discipline is and thus more concentrated. Diversity measures the inverse sum of connecting distances to all other nodes, reflecting the extent to which a node is isolated within the knowledge system: the greater the diversity, the fewer extended connections a discipline has and thus forming more confined small groups in the network. Empirical analyses have demonstrated that concentrated knowledge structures facilitate dissemination of existing knowledge, whereas isolated structure can increase adaptivity to different disciplinary knowledge and facilitate radical innovations to knowledge development through looking from divergent angles (Bodin & Prell, 2011; Foray, 2018; Schot & Geels, 2008).

Based on the differences between the centrality and diversity values, we will classify the knowledge structure of river basins into four types of knowledge structures (Figure 1). They are:

- 1. Ideal structure with high centrality and high diversity. With this structure, the river basin should have high research intensities in core disciplines to provide solid theoretical foundations, while at the same time sufficient cross-disciplinary collaborations to ensure knowledge innovations to address unexpected, emerging river basin management challenges.
- 2. Innovation-inclined structure with high diversity but low centrality, which could have a risk of discipline hollowing-out (marginalization of influence of core discipline). For the river basins with this structure, the connection with core disciplines (centrality) should be strengthened.
- 3. Legacy-driven structure with high centrality and low diversity, which discourages knowledge innovation. In the river basin with this structure, the cross-disciplinary collaborations (diversity) should be strengthened to increase the potential of knowledge pattern transformation against emerging management challenges; and
- 4. Underdeveloped structure with low centrality and low diversity, indicating that the knowledge development is still at its early stage and the knowledge development should be strengthened comprehensively.

Figure 1 Four different knowledge structures based on their structural metrics

Reorganizing the methodology section is needed to make it easier to follow. In its current state, the section starts with definitions, which is fine, but the rest is discussed all around how the data was

processed with methods inserted in the text. It will be better to split up section 2 into three sub-sections "definition", "data" and "methods".

Thank you very much for your comments. We will include a separate section "Framework/Definition", then data sources and data processing, followed by our analysis methods.

The discussion section would be valuable if some thoughts were put in ways to make water research more interdisciplinary than "isolated/centralised knowledge" as defined, for example how gaps identified could contribute to the framings of socio-hydrology, eco-hydrology, etc.

Thank you for your comments. We will revise our key findings based on the results to be updated from our proposed revision of definition of knowledge structure of river basin. Then, we will discuss the implications of our findings on structural deficiency of water resources knowledges to complement to existing findings by professional knowledge and research experience in hydrology. For example, we will assess the development of socio-hydrology and eco-hydrology from the links of hydrology with other relevant disciplines, and recommend if these links should be strengthened and/or new links should be established according to both diversity and centrality at each river basin and all 95 river basins.

The limitation of the paper should be acknowledged in some aspects. To be specific, the data for the knowledge synthesis does not cover grey literature which usually has reported management efforts that are not covered in academic papers. Papers that are not indexed in WoS could have also contributed to the field and be worth acknowledging. The absence of studies is not evidence of the absence of issues/development.

Thank you for your comment. We will discuss the limitations of our study including only journal papers in WoS as you pointed out.

**Specific comments**

**The authors may want to rename the title of the paper as it now does not cover the whole water resources system.**

Thank you for your comment. As we stated in our response to Professor Savenije, we chose river basin as the spatial unit for analysis as it represents the territorial unit of water cycle linking to other cycles of the Earth system (e.g. nutrients, energy, and carbon), which are commonly adopted by researchers to understand the integrated impacts of water use, land use and environmental management (Newson, 2008; Warner et al., 2008). We merged those publications focusing smaller spatial units (e. g. sub-catchment, or wetland or lake into the river basin which they are affiliated with). But we agree that we may have missed the publications on general conceptual/theoretical development without specific spatial links and those publications at global scale. Thus, we will revise our manuscript title as "Gaps of water resources knowledge structure in river basins".

Section 2.1: using the availability of studies to define the knowledge status/gaps, in particular management of rivers, may not be appropriate as management practices could have been implemented to some river basins that have not drawn much academic attention. The absence of studies does not necessarily mean the absence of knowledge development for the basins. The authors should acknowledge its limitations.

Thank you for your comment. We chose academic publications as our data source as it provides systematic documentations of knowledge development across a broad range of disciplines. Large online publication databases enable consistent data retrieval for a long timeframe. However, we do recognise that some river basins may receive fewer academic attention and focus more on practice-driven management. We will acknowledge this limitation in the discussion section.

**The authors used network indicators to measure knowledge connections. However, how the network was built is not well explained. What are nodes and links in the network are not clearly defined in the main text.**

Thank you for your comments. The network connections were established based on the co-occurrence principle. Two disciplines were connected if they were linked to the same key word in a publication; and two management issues were connected if they appeared in the same publication. As a result, the disciplinary network contained different disciplines as nodes, whereas the management issue network contained issues as nodes. The weighting of links were the number of publications. We will include these details in the method section.

Section 2.2: First, using the keywords-based approach to retrieve records sometimes is controversial, because the results are significantly affected by the words selected for data collection. Some justifications of words selection should be added. Second, how groups of concerns were defined (i.e., agricultural irrigation, climate variability, etc.) and how each publication was classified into a specific group will need more explanations. For example, how studies on water policy were distinguished from management, how the overlaps were treated? What about studies of groundwater depletion and agricultural irrigation, were they included in agricultural irrigation or groundwater management? Some examples given may be helpful.

Thank you for your comments. First, key words have widely been used to express the research topics of articles and considered a basic element in understanding the content and structure of disciplinary knowledge (Cheng et al., 2020; Khasseh et al., 2017). In addition, we have used all key words retrieved in the title, abstract and key words sections of each article after natural language processing. We will make these clearer when we revise our manuscript. Second, the nine groups of key words we identified were derived from our data rather than pre-set. Grouping was based on broadly recognised river basin management concerns and the processes underlying them. In addition, as we stated in our response to Professor Savenije, we will re-examine those newly appeared key words in each temporal stage and may categorise them in newly defined groups to more precisely reflect the evolution of management issues. To keep consistency of key words grouping, the two independent coders who did the grouping will be asked to code the key words with any ambiguity thoroughly discussed. Finally, we will include a table in the methods section to list our identified key words groups and give examples for each group. Surface water and groundwater management referred to the general water resources management issue (e.g "water level fluctuations", "drainage"); Water policy refer to the specific policy initiatives and instruments (e.g. "flood management", "integrated management"). All groundwater related issues were grouped into surface water and groundwater management. Agricultural irrigation referred to the specific irrigation methods and techniques. The word co-occurrence in the same publication was used to indicate the relationships between agricultural irrigation and groundwater depletion.

Section 2.2: Which 5 basins, except for St Lawrence River basin, were removed? Justifications should be added to improve the robustness of data. St Lawrence River is a large river basin in North America which connects to the Great Lakes Basin draining all the way up to the Atlantic Ocean. The drainage basin of ST Lawrence River has been ranked 13th largest in the world, providing millions of population and wildlife with water resources. A series of management strategies and actions have been planned since the 1980s, which have made significant progress on the protection of the ecohydrological systems of the basin. https://www.planstlaurent.qc.ca/en/our-history

Thank you for your comment. We removed five river basins: the Lawrence River, the St. Lawrence River, the Red River, the Lena River and the Missouri River as the retrieved river names could not differentiate the rivers (i.e. the Lawrence River and the St. Lawrence River), the retrieved river name did not have clear connection with specific location (i.e. the Red River), and the retrieved river name did not have full-length data during the study period (i.e. the Lena River and the Missouri River). We will acknowledge this in more details in the method section.

Section 3: the total number of publications retrieved was not given in the text. Were all those publications included for the analysis or if any criteria were applied to clean the dataset?

Thank you for your comment. All publications after the filtering of key words were retrieved and used for analysis. A total of 9128 publications from 1970-2017 were finally used for analysis. The publications per year will be attached as supplementary materials in the edited manuscript.

Section 3, line 150: This would indicate that scientists started to focus on/realized synergistic impacts from water quality issues to ecosystems. Less previous studies do not mean that the impacts were not important.

Thank you for your comment. We will revise this sentence as "the interactive impacts between water quality and ecosystem degradations have been a major focus of scientists during our study period".

**Section 4: It would be good to separate discussion and conclusion sections.**

Thank you for your comment. We will separate the discussion section (as briefly outlined above) and the conclusion section (briefly summarise the key findings and implications) in our revised manuscript.

Yours sincerely,

The authors team:

Shuanglei Wu, Yongping Wei, Xuemei Wang

**References**

- Bodin, Ö., & Prell, C. 2011. Social Networks and Natural Resource Management : Uncovering the Social Fabric of Environmental Governance. Cambridge, UNITED KINGDOM: Cambridge University Press.
- Cheng, Q., Wang, J., Lu, W., Huang, Y., & Bu, Y. 2020. Keyword-citation-keyword network: a new perspective of discipline knowledge structure analysis. *Scientometrics*, 124(3), 1923-1943. doi:10.1007/s11192-020-03576-5
- Coccia, M. 2020. The evolution of scientific disciplines in applied sciences: dynamics and empirical properties of experimental physics. *Scientometrics*, *124*(1), 451-487. doi:10.1007/s11192-020-03464-y
- Dampier, W. C. 1944. A shorter history of science: Cambridge University Press.
- Foray, D. 2018. Smart specialization strategies as a case of mission-oriented policy—a case study on the emergence of new policy practices. *Industrial and Corporate Change*, 27(5), 817-832. doi:https://doi.org/10.1093/icc/dty030
- Khasseh, A. A., Soheili, F., Moghaddam, H. S., & Chelak, A. M. 2017. Intellectual structure of knowledge in iMetrics: A co-word analysis. *Information Processing & Management*, 53(3), 705-720. doi:https://doi.org/10.1016/j.ipm.2017.02.001
- Newson, M. 2008. Land, water and development: sustainable and adaptive management of rivers: Routledge.
- Schot, J., & Geels, F. W. 2008. Strategic niche management and sustainable innovation journeys: theory, findings, research agenda, and policy. *Technology Analysis & Strategic Management*, 20(5), 537-554. doi:10.1080/09537320802292651
- Shi, F., Foster, J. G., & Evans, J. A. 2015. Weaving the fabric of science: Dynamic network models of science's unfolding structure. *Social Networks*, 43, 73-85. doi:https://doi.org/10.1016/j.socnet.2015.02.006
- Warner, J., Wester, P., & Bolding, A. 2008. Going with the flow: river basins as the natural units for water management? *Water Policy*, 10(S2), 121-138. doi:10.2166/wp.2008.210

---

## Author Response (AR1)

**Response Letter**

Dear Prof. Savenije, Dr. Xu, and Dr. Bogaard,

Thank you very much for your time and effort in reviewing the manuscript initially titled "Global water resources knowledge gaps".

We have revised the manuscript title to "**Structural gaps of water resources knowledge in global river basins**" to better reflect the focus of our study.

We really appreciate all of your insightful comments. To fully address them, please see the revised version of the manuscript (a clear version and a track-changed version). As the manuscript has been substantially rewritten, we will not include all details but refer to specific sections of the manuscript by line numbers in this point-by-point responses to each of your comments below:

**COMMENTS FROM PROF. SAVENIJE:**

*The authors have analysed papers categorised in the WoS under Water Resources and limited the analysis to articles that deal with river basins or catchments in a broad sense. They then looked for connections between disciplinary fields of WRM and analysed the connections between these fields and how they developed over time. They then classified the patterns of interconnection, or lack thereof into knowledge structures with the following names: Isolated, Innovative-inclined, Legacy-inclined and Centralised. To me, these classifications have hardly any explanatory power. I have gone through the description several times, but I fail to see what these terms actually mean or imply in relation to WRM. I can't see whether they have a positive or negative connotation. To me, Isolated and Centralised sounds rather negative; Innovative-inclined sounds positive; and Legacy-inclined may be both positive or negative, depending on one's perspective. That in recent years more basin studies are legacy-inclined may be evidenced by the data, but I have difficulty to see what it means.*

Thank you for your comments. We have revised our method and data section to clarify the knowledge structure metrics and their significances (**Line 60-100**) and the revised knowledge structure types have been graphically illustrated in **Figure 1** of the revised manuscript.

*I do understand that this is a data-mining exercise, and that the authors did not necessarily familiarize themselves with the field of Water Resources Management and its development over time. I recommend looking at the paper on "Evolving water science in the anthropocene" (https://hess.copernicus.org/articles/18/319/2014/) and the huge body of papers that have recently been published under the IAHS research initiative "Panta Rhei", e.g. "Global perspectives on hydrology, society and change", Hydrological Sciences Journal, 61:7, 1174-1191, (DOI:10.1080/02626667.2016.1159308).*

Thank you very much for your comments. We have fully revised the introduction section of our manuscript to clarify the motivation of this study and review existing literature gaps that

can be addressed by our approach. The detailed introduction can be found in **Line 20-60** of the revised manuscript.

In addition, we have also revised the discussion section of our manuscript with the recommended references and additional ones to discuss the implications of our findings from the perspective of structurally complementing existing professional knowledge and research experiences (**Line 280-365**).

*By taking river basins as the entree point, I fear the authors have missed a huge body of conceptual and global research. Not all WR research is done at river basin level. Much happens at the global scale, national scale, policy scale or conceptually.*

Thank you for your comments. We have provided more details on the justifications why river basin was chosen as the spatial unit for analysis in the method section (**Line 100-115).** In addition, as explained in the method section, we merged those publications focusing smaller spatial units (e.g. sub-catchment, or wetland or lake into the river basin which they are affiliated with). We have also recognised the limitations in the river basin scale chosen by this study in the discussion section (**Line 355-360**), which may miss general conceptual/theoretical development without specific spatial links and those studies on global scales.

*By choosing to analyse traditional disciplinary fields, such as: Agricultural irrigation; Erosion and sedimentation; Water pollution and treatment; Surface water and groundwater management; Ecological degradation; Droughts and floods; Climate variability and change; the results obtained are hardly pointing towards stronger societal linkages. I miss emerging new fields, such as: demand management; decentralisation; participation; international water law; .... and new technologies such as Remote Sensing, New observation technology, Global modelling, Artificial intelligence, .... If you look for traditional terms, you are bound to find traditional results.*

Thank you for your comments. We have clarified in the method and data section how to retrieve and group key words with the classical bibliometric approach (**Line 120-140**). In addition, we have also re-examined and highlighted those newly appeared key words in each temporal stage to precisely reflect the evolution of management issues in the result section **(Line 225-250).** We have also included a table in **Appendix A** that lists the identified management issue groups and give examples for key words classified in each group. It should also be noted that the computer-based text-mining technique was adopted, so some key words might be missed.

*Section 4. Discussion and Conclusion, is hardly a discussion. It is rather a set of three recommendations where certain lines of research are "encouraged": 1) investigation of "new" river basin phenomena; 2) spatial diversity of Water Resources research; and 3) strengthening collaborations with social sciences. These are rather obvious and general recommendations and hardly a discussion. The conclusion that "the stationarity of the water resources knowledge system persist" is not supported by the large body of work that is recently being produced as a result of and as part of the "Panta Rhei" initiative. This large body of work is hard to detect if one constrains oneself to the 95 most studied river basins in the world and the connections to traditional fields.*

Thank you for your comments. We have substantially revised our discussion section based on the results and draw the conclusion from the perspective of knowledge structure of river basin. Detailed discussions can be found in **Line 280-370**.

**COMMENTS FROM DR. XU:**

*General comments:*

*This work aimed to discover knowledge gaps in water resources research at the river basin scale through looking into the knowledge structure and disciplinary connections over time. The starting point of this paper is very interesting and the topic is important as river management and governance are highly fragmented. Generalizing knowledge patterns for research and management practices at the basin scale is challenging but should be done. Identification of knowledge gaps through investigating the knowledge structure is an innovative approach. Tracing the knowledge development patterns could also help identify gaps between science and policy, which is critical for the knowledge mobilization that promotes science-based decision-making for water systems. The synthesis of such fragmented knowledge would be benefited from large data analytics such as text mining approaches and content analysis. Text mining is an efficient way for the synthesis of knowledge which otherwise will be buried in the large number of texts.*

*This paper used academic literature obtained from the Web of Science as the main source and made use of a text-mining approach to extract key terms from the literature. The authors then used two indicators (degree and closeness) to measure connections among knowledge domains defined in this study. Overall, the methodology is designed in a reasonable manner and discussions are fair. However, some revisions are required to make it more readable and informative.*

Thank you very much for your positive comments.

*Knowledge structure is a keyword of the paper and it is a cognitive concept/science which needs to be carefully defined. It has been well defined in many other disciplines such as education, psychology, etc. What does it mean in water science at the basin scale?*

Thank you for your comment. We have clarified the definition of knowledge in the context of the water resources discipline in this study in the introduction (**Line 25-50**) and method section 2.1 (**Line 60-70**).

*It will be beneficial for the paper to list definitions of terms in a table (i.e., limited development, isolated development, innovative-inclined development, legacy-inclined development, centralised development). As these terms are not commonly used in the context of water sciences, nor is it in knowledge evolution, one might need to go back to read definitions a few times before he/she could understand and remember them. If they are new to the field, the authors should make them clearer to be understood. A diagram that distinguishes them from each other would be helpful as they are now ambiguous. Alternatively, the authors may need*

*to rephrase them into terms that are more common (e.g., "lack of knowledge", "disciplinary", "multidisciplinary", "interdisciplinary", "transdisciplinary", etc. Tress et al., 2005. Clarifying integrative research concepts in landscape ecology. Landscape Ecology, 20, 247-493).*

Thank you for your comments. We have reorganised our classification of knowledge structure into four types and clarify their implications on water resources knowledge development (**Line 60-100**) and graphically in **Figure 1** of the revised manuscript.

*Reorganizing the methodology section is needed to make it easier to follow. In its current state, the section starts with definitions, which is fine, but the rest is discussed all around how the data was processed with methods inserted in the text. It will be better to split up section 2 into three sub-sections "definition", "data" and "methods".*

Thank you very much for your comments. We have reorganised the methods and data section to ensure a more logical flow (**Line 60-165**). We added separate sub-sections in the method and data section that first defined the structure of the water resources knowledge system (2.1), followed by data sources (2.2), key words analysis (2.3), knowledge network analysis (2.4) and temporal periods division (2.5).

*The discussion section would be valuable if some thoughts were put in ways to make water research more interdisciplinary than "isolated/centralised knowledge" as defined, for example how gaps identified could contribute to the framings of socio-hydrology, eco-hydrology, etc.*

Thank you for your comments. We have revised our discussion section based on the key findings from our results from the perspective of knowledge structures of river basins. Detailed discussions can be found in **Line 280-370**.

*The limitation of the paper should be acknowledged in some aspects. To be specific, the data for the knowledge synthesis does not cover grey literature which usually has reported management efforts that are not covered in academic papers. Papers that are not indexed in WoS could have also contributed to the field and be worth acknowledging. The absence of studies is not evidence of the absence of issues/development.*

Thank you for your comment. We have discussed the limitations of our study in **Line 355-365**, in recognising the limited study scope of using only journal papers in WoS as well as other limitations.

*Specific comments:*

*The authors may want to rename the title of the paper as it now does not cover the whole water resources system.*

Thank you for your comments. We have revised our manuscript title as "Structural gaps of water resources knowledge in global river basins" to better reflect the focus of our study.

We provided details on the justifications about choosing river basin as the spatial unit for analysis in the method section (**Line 100-120**). We have also discussed the limitation of only choosing river basins in **Line 355-365**.

*Section 2.1: using the availability of studies to define the knowledge status/gaps, in particular management of rivers, may not be appropriate as management practices could have been implemented to some river basins that have not drawn much academic attention. The absence of studies does not necessarily mean the absence of knowledge development for the basins. The authors should acknowledge its limitations.*

Thank you for your comment. We chose academic publications as our data source as it provides systemic documentations of knowledge development across a broad range of disciplines. Large online publication databases enable consistent data retrieval for a long timeframe (**Line 100-115**). However, we do acknowledge that some river basins may receive fewer academic attention and focus more on practice-driven management. This limitation has been discussed in **Line 355-365**.

*The authors used network indicators to measure knowledge connections. However, how the network was built is not well explained. What are nodes and links in the network are not clearly defined in the main text.*

Thank you for your comments. We have added a separate sub-section in the method and data section to clarify the establishment of knowledge network (**Line 140-155**).

*Section 2.2: First, using the keywords-based approach to retrieve records sometimes is controversial, because the results are significantly affected by the words selected for data collection. Some justifications of words selection should be added. Second, how groups of concerns were defined (i.e., agricultural irrigation, climate variability, etc.) and how each publication was classified into a specific group will need more explanations. For example, how studies on water policy were distinguished from management, how the overlaps were treated? What about studies of groundwater depletion and agricultural irrigation, were they included in agricultural irrigation or groundwater management? Some examples given may be helpful.*

Thank you for your comments. We have clarified in the method and data section about how to retrieve and group key words using the classical bibliometric approach (**Line 120-140**). In addition, we have also re-examined and highlighted those newly appeared key words in each temporal stage to precisely reflect the evolution of management issues in the result section **(Line 225-250).** Finally, we have included a table in **Appendix A** that lists the identified management issue groups and give examples for key words classified in each group (**Line 375).**

*Section 2.2: Which 5 basins, except for St Lawrence River basin, were removed? Justifications should be added to improve the robustness of data. St Lawrence River is a large river basin in*

*North America which connects to the Great Lakes Basin draining all the way up to the Atlantic Ocean. The drainage basin of ST Lawrence River has been ranked 13th largest in the world, providing millions of population and wildlife with water resources. A series of management strategies and actions have been planned since the 1980s, which have made significant progress on the protection of the ecohydrological systems of the basin. https://www.planstlaurent.qc.ca/en/our-history*

Thank you for your comment. We have explained the reason why the five river basins including the ST Lawrence River were removed in the method section (**Line 100-115**).

*Section 3: the total number of publications retrieved was not given in the text. Were all those publications included for the analysis or if any criteria were applied to clean the dataset?*

Thank you for your comment. All publications after the filtering of key words were included and used for analysis. A total of 9128 publications from 1970-2017 were finally used for analysis. The details were added in **Line 165-170**.

*Section 3, line 150: This would indicate that scientists started to focus on/realized synergistic impacts from water quality issues to ecosystems. Less previous studies do not mean that the impacts were not important.*

Thank you for your comment. We have removed the results about synergistic impacts of management issues including this sentence.

*Section 4: It would be good to separate discussion and conclusion sections.*

Thank you for your comment. We have separated the discussion section (**Line 280-365**) and the conclusion section (**Line 365-370**) in our revised manuscript.

Yours sincerely,

The authors team:

Shuanglei Wu, Yongping Wei, Xuemei Wang

---

## Author Response (AR2)

**Response Letter**

Dear referees and editor,

Thank you very much for your time and effort in again reviewing the manuscript titled "**Structural gaps of water resources knowledge in global river basins**".

We really appreciate all of your insightful comments. To fully address them, please see the revised version of the manuscript (a clear version and a checked changed version). We have also provided point-by-point responses to each of your comments below:

**COMMENTS FROM PROF. SAVENIJE:**

Thank you for your positive feedbacks to this revised manuscript and very constructive comments in the previous version.

**COMMENTS FROM ANONYMOUS REFEREE #2:**

Thanks for the manuscript. I have enjoyed reading this paper which has been improved compared to the previous manuscript. The authors have improved their discussions in a way that engages with other interdisciplinary subfields in water sciences, e.g., socio-hydrology, providing insights for the interdisciplinary studies for river basin studies.

In my opinion, however, some minor revisions are needed. 1) careful proofreading will be beneficial for the paper. There are typos and grammatical mistakes (e.g., first sentence in 2.1, "on this field" on pp3, "...number of rivers basins" on pp9, etc.); 2)Please clarify what "data issues" means on pp4.

Thank you for your positive comments on this revised manuscript. We have:

1) Carefully proofread the full manuscript and corrected the grammatical errors, including changing "on this field" on pp3 to "in this field", and changing "number of rivers basins" on pp9 to "number of river basins". More minor corrections are annotated in the author tracked changed version.
2) The "data issues" on pp4 refer to river basins with incomplete data (i.e., missing and/or mismatches key words and disciplines) and river basins identified with ambiguous names (e.g., same names but different spatial locations). We have added additional explanation in Line 115.

Yours sincerely,

The authors team:

Shuanglei Wu, Yongping Wei, Xuemei Wang